# Measuring the Sustainable Development of Marine Economy Based on the Entropy Value Method: A Case Study in the Yangtze River Delta, China

**Xiaowei Ni and Yongbo Quan \***

School of Economics and Management, Zhejiang Ocean University, Zhoushan 316022, China
\* Correspondence: quanyongbo@zjou.edu.cn

**Abstract:** The rapid growth of the marine economy has provided a strong impetus for the economic development of the Yangtze River Delta region, but it has also posed serious challenges to the ecological environment of the marine watersheds in which it is located. To investigate how to promote the sustainable development of the marine economy in the Yangtze River Delta region, this study used relevant statistical data from 2009 to 2019, combined with four factors—marine economy, marine resources, ecological environment, and scientific and technological innovation—to build an evaluation index system for the sustainable development of the marine economy and employed the entropy value method to conduct a deeper investigation. It was found that there are problems in the sustainable development of the Yangtze River Delta marine economy, such as imperfect marine industry structure, significant differences in resource allocation, insufficient support capacity of marine science and technology innovation, and insufficient ecological environmental protection. Therefore, the study suggests promoting regional collaboration in the Yangtze River Delta, improving technological innovation, and enhancing environmental protection to support the sustainable development of the regional marine economy.

**Keywords:** marine economy; sustainability; indicator system; Yangtze River Delta; entropy value method





## 1. Introduction

With the dramatic increase in population and growth in economies, developing countries are facing problems of the depletion of terrestrial resources, space constraints, and environmental degradation, which seriously affect the process of sustainable development [1,2]. The growth of the marine economy has provided significant development opportunities for coastal countries and those with interests in marine industries [3]. However, with increasing industrialization, coastal areas are also confronted with various environmental challenges, including the discharge of industrial wastewater, domestic sewage, solid waste, and marine oil spills [4]. The Yangtze River Delta region is a crucial hub in China's drive to build a "strong ocean country," contributing a considerable portion of the nation's gross marine product. Yet, it faces severe environmental pollution issues, particularly in the Yellow Sea and East China Sea basin, accounting for 79.48% of China's jurisdictional waters. Thus, while promoting the development of the marine economy in the region, safeguarding marine ecology and resources is essential to ensure sustainable growth and advance the prospects of the regional marine economy.

The majority of studies on the sustainable development of the marine economy have concentrated on the current state of the marine economy as well as theoretical studies and influencing factors of sustainable development. The marine economy is a complex and dynamic system influenced by various factors such as pollution, overfishing, and coastal development [5]. Environmental pollution is a major factor because it can pollute marine resources, harm the health of marine ecosystems, and impair coastal communities' ability to adapt to change [6,7]. Overfishing is a major contributor, as it can lead to fish

stock depletion and the destruction of marine habitats [8]. Finally, coastal development has the potential to devastate the marine environment and displace coastal communities. Several strategies have been proposed to address these challenges for the sustainable development of the marine economy. These strategies include the creation of marine protected areas [9,10], the adoption of sustainable fishing practices [11], pollution reduction, and the promotion of environmentally friendly coastal development [12]. Some academics believe that implementing ocean governance at the governmental level is a critical step toward achieving sustainable ocean economy development [13]. Adapting and optimizing existing governance systems can improve marine environmental protection from various perspectives [14,15]; creating new sectors to implement integrated ocean management can also help improve overall marine economic development [16,17].

The marine economy is becoming increasingly important in the social development and improvement of living standards in coastal areas and regional economic and social development [18]. However, due to the rapid development of the marine economy, the marine ecosystem and environment that form the foundation of the marine economy's development are deteriorating and being lost, resulting in many issues such as the decline of the quality of the marine environment, the destruction of the marine ecosystem, and the continuous weakening of ecological capacity [19]. In light of the current development quandary, the transformation of marine economic growth is an important path for the long-term development of the marine economy [20]. At the same time, many studies have explored the impact of the ecological environment on the development of the marine economy from the regional marine ecological environment [21,22], arguing that the marine economy and marine environment can interact positively [23].

The input of marine resources and technological innovation constitutes an essential element of the marine economic system. Some scholars believe that the use and input of marine resources are critical to the development of regional marine economies, that its impact on the efficiency of marine economic development varies over time [24], and that improving energy development and utilization efficiency can help achieve sustainable development goals [25]. Several other scholars have investigated the relationship between the ocean economy and the environment from the standpoint of the ocean's carrying capacity [26], and they have made insightful policy recommendations to achieve sustainable ocean development [27]. On the other hand, excessive marine resource inputs will result in resource congestion, limiting the efficiency of economic output and sustainable development [28]. Marine technology innovation is a critical factor in the regional economy's healthy and sustainable development [29,30], and it is the primary mechanism for achieving sustainable development. Through innovative technology, marine science and technology can strengthen related enterprises and industries [31,32], which is the endogenous driving force for the rapid development of the marine economy, and better monitor the marine environment [33].

In addition, different scholars use different methods to measure the weight of indicators of sustainable development. Li et al. used the DPSIR model to combine economic and ecological impacts to construct a prediction system for the ecological sustainability of the Bohai Rim in China from five aspects: drivers, pressures, states, effects, and responses [34]. Lin used the coupled coordination degree model and information entropy method to establish an index system to measure the interrelationship between the marine economy and the ecological environment to achieve sustainable development [35]. Data envelopment analysis (DEA) was used by Wen et al. to assess the coordination and coupled development of marine economies and ecosystems [36]. As an objective weight assignment method, the entropy method is widely used in the measurement of sustainable development because it is not influenced by subjective factors and has high reliability. It is commonly used in the measurement of sustainable development. Most research related to sustainable development uses this method to create new indicator systems to measure more comprehensive abstract concepts. He et al. used the entropy method to gauge the state of clean energy development in various nations [37]. Gong et al. constructed a system of indicators to

compare the level of sustainable urban development in five dimensions: social, economic, environmental, resource, and technological [38]. Jin et al. presented a new country sustainable development indicator to improve on the widely used Human Development Index [39]. Sun et al. comprehensively analyzed and evaluated regional economic development using the entropy method in response to regional environmental problems and national policy goals of the green development of ecological and environmental construction [40]. These studies demonstrate the broad applicability of the method.

Previous studies have focused on specific sectors of the marine economy, or on individual factors, rather than taking a holistic approach that considers the impact of multiple factors on the sustainable development of the marine economy. Previous studies lacked comprehensive standardized data on the marine economy, which made it difficult to accurately assess its current state and sustainable development potential, and the assessment methods used were subject to issues of subjectivity and internal factors affecting the sustainable development system. Based on this, this study has several innovations and contributions. First, this study introduces several factors such as the current state of the marine economy, marine environment, ecological resources, and scientific and technological innovation, and establishes a more complete and comprehensive evaluation system that focuses more on the quality of development from the perspective of sustainable development. Secondly, the relationship between the current situation of the marine economy, marine environment, ecological resources, and the scientific and technological innovation and sustainable development of the marine economy is analyzed using the entropy value method, taking into account the objectivity and completeness of the evaluation indexes. Again, the Yangtze River Delta region is an important fulcrum of China's marine economic development, but few scholars have chosen it as a research area. This study explores the current situation of the sustainable development of the marine economy in the Yangtze River Delta region and the regional differences, and this study will provide referenceable insights for coastal areas on how to properly promote the sustainable development of the marine economy.

## 2. Construction of the Evaluation Index System for the Sustainability of the Marine Economy

The sustainable growth of the marine economy is an important foundation for supporting socio-economic development, as well as for the optimal allocation and development of marine resources [41]. Based on an understanding of the meaning of sustainable development in the context of the United Nations Sustainable Development Goals and China's ocean power strategy, three coastal provinces in the Yangtze River Delta are used as examples in this paper. Regarding the research of existing scholars [42], this study takes into account the scientific validity, comprehensiveness, and data availability of index selection. It constructs a sustainable development evaluation index system of marine economy containing 14 specific indicators from four dimensions: marine economy, marine resources, ecological environment, and scientific and technological innovation (Table 1).

Marine economy: Accelerating the growth of the marine economy is an inherent requirement for regional development and a powerful economic growth engine. The development of the marine economy in coastal areas affects the sustainable development potential of the region [43], and it needs to form a reasonable and diversified industrial structure [44] and maintain long-term vitality. As a result, in order to assess the level of development of the marine economy, not only the scale and speed of the marine economy but also the quality of marine economic development must be considered [45]. In this paper, the proportion of gross marine product to coastal area gross product is used to reflect the scale of marine economic development in Yangtze River Delta provinces [46]. The added value of major marine industries and marine-related industries is used to reflect the speed of marine economic development. Based on the theory of social reproduction, crude growth only brings short-term benefits and is not conducive to the sustainable development of the marine economy. The structure of the marine industry reflects a greater extent the growth

characteristics of the marine economy in terms of quality, so we used the proportion of tertiary industry to observe the rationality of the industrial structure [47].

**Table 1.** Evaluation system of the sustainable development level of the Yangtze River Delta marine economy.

| Primary Indicator | Secondary Indicator | Unit | Nature |
|---|---|---|---|
| Marine economy | The proportion of marine GDP to regional GDP | % | Positive |
| | Value-added of major marine industries | Billion yuan | Positive |
| | Value-added of marine-related industries | Billion yuan | Positive |
| | The proportion of marine tertiary industry | % | Positive |
| Marine resources | Per capita water resources | Cubic meters per person | Positive |
| | Production of marine products | Ton | Positive |
| | Number of berths for production above 10,000 tons in ports | - | Positive |
| Ecological environment | Industrial wastewater emissions | Billion tons | Negative |
| | Industrial solid waste emissions | Million tons | Negative |
| | Investment in pollution control as a proportion of GDP | % | Positive |
| Technological innovation | Number of marine scientific research institutions | - | Positive |
| | Number of marine scientific researchers | - | Positive |
| | Number of scientific papers in marine research institutions | - | Positive |
| | Number of invention patents owned by marine research institutions | - | Positive |

Marine resources: As an important input factor of the marine industry system [48], marine resources have biological resources value, habitat resources value, supply service value, and species diversity maintenance service value. Although Shanghai, Jiangsu, and Zhejiang Provinces are positioned in the same sea area, each province has its own resources involved in the development of the marine economy. To ensure the comparability of data indicators, three secondary indicators were selected to measure marine habitat resources, biological resources, and service levels: water resources per capita, seawater products production, and the number of berths for production in ports over 10,000 tons [49].

Ecological environment: Due to the frequent use of marine space, the cumulative impact on the marine ecosystem is increasing [50]. In the process of accelerating maritime economic development, we must focus on the synergy between economic growth and environmental protection, and we must be aware of the adverse effects of environmental degradation on economic growth. The efficacy of pollutants from terrestrial sources and their treatment is important for the sustainable development of oceans. We chose the indicators of industrial wastewater emissions and industrial solid waste emissions to represent the degree of ocean pollution [51,52]. The share of pollution control investment in regional GDP was used to measure the importance and effective response of each region to environmental protection [53].

Technological innovation: Marine technology innovation can promote the rapid development of related marine industries and is a booster to promote the rapid growth of the marine economy [54]. Therefore, this paper selected the number of marine scientific research institutions and scientific researchers as the input of technological innovation and used the number of scientific papers and invention patents of marine scientific research institutions to measure the effect of scientific innovation output [55].

## 3. Data and Research Method

### 3.1. Study Area

The Yangtze River Delta refers to the Yangtze River's downstream region, which is one of China's regions with the most marine economic development, the most coastal

development, and the greatest concentration and abundance of marine resources. The region includes three coastal provinces: Shanghai, Jiangsu, and Zhejiang, all of which contribute significantly to China's marine economy.

Shanghai achieved a marine GDP of 103.663 billion RMB in 2021, ranking fourth in China. It has the most rapidly developing coastal tourism and transportation industries. Jiangsu's coastal mudflat area accounts for about a quarter of the total mudflat area in China, with rich harbor and navigation, land, and biological resources. Zhejiang Province's coastline is the longest in the country, equivalent to 2.6 times that of the province's land area. At the same time, Zhejiang also has many islands and more than 700 km of deep-water coastline. The proportion of its marine GDP to the regional GDP is higher than the national average.

### 3.2. Data Sources

Since most of the official data from the last three years have not been published for most indicators in this paper, data for 2009–2019 were selected to be analyzed to conduct a more comprehensive study. The data on the marine economy, marine resources, and science and technology innovation were obtained mainly from China Marine Statistical Yearbooks (2010–2017) (https://data.cnki.net/, accessed on 27 March 2023) and China Marine Economic Statistical Yearbooks (2010–2020) (https://data.cnki.net/, accessed on 27 March 2023), and the data on the environmental pollution in this paper were obtained from China Environmental Statistical Yearbooks (2010–2020) (https://data.cnki.net/, accessed on 27 March 2023) and China Marine Environmental Status Bulletins (2010–2020) (https://www.mee.gov.cn/hjzl/sthjzk/jagb/, accessed on 27 March 2023). Missing data were completed by data from the statistical yearbooks of each region.

### 3.3. Method

In order to reflect the level of the sustainable development of the marine economy, a comprehensive evaluation requires multiple indicators. The determination of indicator weights, as an important part of the model evaluation, will directly affect the evaluation results. Hence, this paper proposes a method for measuring and analyzing the sustainability of the marine economy through the use of entropy values. The theoretical foundation of the entropy method rests on the concept of information entropy, which was originally introduced by Claude Shannon in 1948 [56]. The entropy method is a decision-making model that utilizes the principle of entropy to measure the uncertainty and information content of a set of data. Specifically, information is a measure of the degree of order in a system, while entropy is a measure of the degree of disorder in a system. Therefore, as the information entropy of indicators decreases, the information provided by the indicators increases. In the context of the comprehensive evaluation, the role and weight of indicators with lower information entropy become more significant [57,58]. Furthermore, compared to other comprehensive evaluation methods, the entropy method is an objective weighting method that calculates the information entropy of indicators and determines the weight of each indicator based on its relative changes, thereby effectively avoiding subjective influences [59]. As a result, the weight results of the entropy method possess a high reference value. The specific calculation steps are as follows:

This study includes T evaluation indicators to measure the level of sustainable development of the regional marine economy in a total of m provinces for n years. Due to the differences in positive and negative orientations among the indicators, and to avoid the influence of zero values on the calculation of information entropy, the data need to be standardized first, as follows:

If it is a positive indicator,

$$y'_{hij} = \frac{X_{hij} - minX_j}{maxX_j - minX_j} \times 99 + 1 \tag{1}$$

If it is a negative indicator,

$$y'_{hij} = \frac{maxX_j - X_{hij}}{maxX_j - minX_j} \times 99 + 1 \tag{2}$$

where $h$ is the year, $i$ is the province, $j$ is the index, and $X_{hij}$ is denoted as the $j$ index value of the $i$ province in the $h$ year; $y'_{hij}$ represents the standardized value; and $minX_j$ and $maxX_j$ denote the minimum and maximum values of $X_j$.

Thereafter, calculate the proportion of province $i$:

$$Y_{hij} = \frac{y'_{hij}}{\sum_{i=1}^{mn} y'_{hij}} \tag{3}$$

where $Y_{hij}$ is the weight of the $i$ indicator value in $h$ year under the $j$ indicator.

Furthermore, the information entropy value for the $j$ indicator is:

$$e_j = -\frac{1}{lnmn} \sum_{i=1}^{mn} Y_{hij} \ln Y_{hij} \tag{4}$$

In this step, we need to make ensure that $0 \leq e_j \leq 1$.

Then the information entropy difference degree $d_j$ is calculated for the $j$th indicator, as follows:

$$d_j = 1 - e_j \tag{5}$$

where $d_j$ stands for the coefficient of variation, indicating the degree of inconsistency in the contribution of each item to the $j$th indicator; in it, the higher the value, the more important it is.

Thereafter, the outcome of Equation (5) is then taken into Equation (6) to calculate the weights of each indicator:

$$W_j = \frac{d_j}{\sum_{j=1}^{T} d_j} \tag{6}$$

where $W_j$ is the $j$th indicator weight, and $\sum_{i=1}^{T} W_j = 1$.

Then, calculate the comprehensive score of the sustainable development level of the marine economy $F_i$, as follows:

$$F_i = \sum_{j=1}^{T} W_j y'_{hij} \tag{7}$$

## 4. Analysis

### 4.1. Measurement Results of Index System

Based on the panel data of three provinces of the Yangtze River Delta from 2009 to 2019, the entropy method was used to calculate the weights of the indicators of marine economy, marine resources, ecological environment, and scientific and technological innovation, and the results are shown in Table 2.

Based on the panel data of 30 provinces and cities in three provinces of the Yangtze River Delta from 2009 to 2019, the entropy value method was used to calculate the weights of the marine economy, marine resources, ecological environment, and science and technology innovation indicators, and the results are shown in Table 1.

The four indicators affecting the sustainable development of the marine economy are marine resources, science and technology innovation, marine economy, and ecological environment, with weights of 0.3253, 0.3143, 0.2473, and 0.1132, respectively. The results show that marine resources and science and technology innovation are important driving factors for the sustainable development of the marine economy. From the secondary indicators, the number of invention patents owned by marine research institutions carries the highest weight of 0.1385, while other indicators with higher weight included water

resources per capita (0.1295), the production of seawater products (0.1287), and marine GDP as a percentage of regional GDP (0.1149). As can be seen, the output of marine scientific research, marine habitat resources, marine biological resources, and the scale of the marine economy in each region are the main driving factors for the sustainable development of the marine economy in the Yangtze River Delta region.

**Table 2.** The index weights of the sustainable development level of the Yangtze River Delta marine economy.

| Primary Indicator | Weights | Secondary Indicator | Weights |
|---|---|---|---|
| Marine economy | 0.2473 | The proportion of marine GDP to regional GDP | 0.1149 |
| | | Value-added of major marine industries | 0.0318 |
| | | Value-added of marine-related industries | 0.0300 |
| | | The proportion of marine tertiary industry | 0.0706 |
| Marine resources | 0.3253 | Per capita water resources | 0.1295 |
| | | Production of marine products | 0.1287 |
| | | Number of berths for production above 10,000 tons in ports | 0.0670 |
| Ecological environment | 0.1132 | Industrial wastewater emissions | 0.0477 |
| | | Industrial solid waste emissions | 0.0159 |
| | | Investment in pollution control as a proportion of GDP | 0.0495 |
| Technological innovation | 0.3143 | Number of marine scientific research institutions | 0.0458 |
| | | Number of marine scientific researchers | 0.0680 |
| | | Number of scientific papers in marine research institutions | 0.0620 |
| | | Number of invention patents owned by marine research institutions | 0.1385 |

### 4.2. Analysis of the Marine Economy

From the analysis of time evolution, the scores of marine economy indicators in each province of the Yangtze River Delta showed a slow upward trend, with a large difference in level but a similar overall trend (Figure 1). In 2014, Shanghai had a significant decrease in the proportion of marine GDP due to the adjustment of production structure, which made the marine economy score show a downward trend. With the construction of Shanghai's free trade zone, Shanghai's marine economy continued to develop, and major marine industries and related industries both developed rapidly with policy support. Jiangsu and Zhejiang, on the other hand, had been on a steady upward trend. The new round of the coronavirus pandemic that started in 2019 inevitably caused the regional marine economy to be negatively affected, leading to a decline in the added value of the national marine industry, while the related marine tertiary industry, which is extremely sensitive to emergencies, was most affected. From the analysis of regional differences in Shanghai, as the core area of the Yangtze River Delta region, the development of the marine economy in this region has greater advantages [60]. Influenced by the national policy strategy and international city positioning, Shanghai scored much higher than other provinces, among which Zhejiang and Jiangsu showed an almost parallel trend in the level of marine economic development. The gap between Zhejiang and Jiangsu gradually became smaller from 2009 to 2011 and tended to be parallel from 2013 to 2019. Due to the huge differences in the scale and industrial structure of the marine economy in each region, the gap between Shanghai's and Zhejiang and Jiangsu's marine economies tended to expand gradually after 2016.

### 4.3. Analysis of Marine Resources

The amount of marine resources varies greatly among the provinces and cities in the Yangtze River Delta, and the allocation and use of each resource have their own focus. In terms of time evolution, the scores of Zhejiang Province fluctuated significantly, while the scores of Shanghai and Jiangsu both changed in a more moderate trend (Figure 2). The score of Zhejiang Province fluctuated mainly due to the per capita water resources and seawater production indicators, reaching a very small value in 2011 and then fluctuating upwards. In

2015, with the port integration initiative, Zhejiang Province integrated the ports of Ningbo and Zhoushan and further improved the level of port construction. This led to a very large score of 30.91 for Zhejiang Province's marine resources in 2015 and then showed a slight decline until 2019 when it gradually showed an upward trend. The fluctuation of Shanghai was not obvious, while Jiangsu Province reached a great value of 9.19 in 2016 due to the sudden increase in water resources per capita and showed a decreasing trend thereafter. From the analysis of regional differences, Zhejiang Province obtained the leading score in the development level of marine resources [61], while Jiangsu and Shanghai's scores were more concentrated, and the gap between them gradually became smaller. Based on the important role of marine resources, provinces and cities should make full use of marine resources around the world, and at the same time can strengthen the circulation of resources through regional cooperation to reduce the differences.

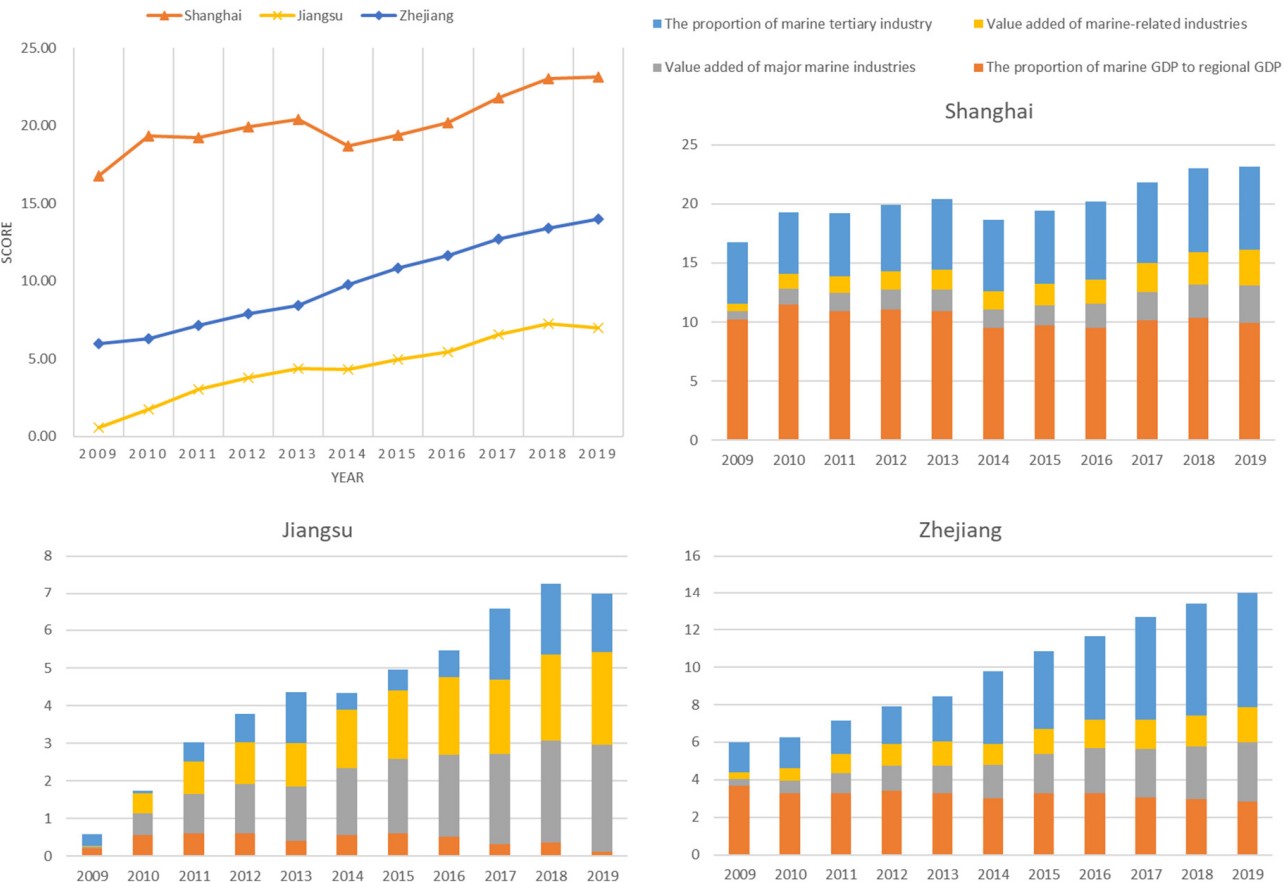

**Figure 1.** Yangtze River Delta marine economic indicator scores from 2009 to 2019.

### 4.4. Ecological Environment Analysis

Analyzed from the perspective of time evolution, Shanghai and Zhejiang showed large fluctuations in the ecological environment (Figure 3). Shanghai declined in an h-shape, with the rate of decline in the score slowing down between 2009 and 2012. In 2012, the State Council put forward the strategy of building a strong marine country, emphasizing the policy focus on coordinating marine ecological protection with the development of the marine economy, which led to a greater focus on marine ecological protection. The level of ecological development increased from 2012 to 2014 due to the significant decrease in two negative indicators of wastewater and solid waste emissions. However, due to the gradual reduction in the investment ratio of pollution control, Shanghai reached the lowest value of 7.16 in 2019 and developed a downward trend. Zhejiang Province showed a fluctuating rise until 2016, rising to a great value of 7.50 and then falling sharply due to the decrease in the investment ratio of pollution control, but showing an upward trend again

in 2019. Jiangsu Province showed a small change due to the decrease in pollution control investment, which brought it back down to the 2014 level in 2018. Although Jiangsu still ranks among the lowest coastal provinces, it was generally on an upward trend. Analyzing the regional differences, Shanghai scored consistently in the lead. As provinces and cities fluctuated, the change in the disparity between regions also fluctuated. Since 2016, there was a trend of narrowing the gap between provinces and cities. In summary, it can be seen that focusing on enhancing marine environmental protection should pay attention to reducing the discharge of wastewater, increasing investment in pollution control, taking the road of sustainable development, and achieving a harmonious coexistence between humans and nature.

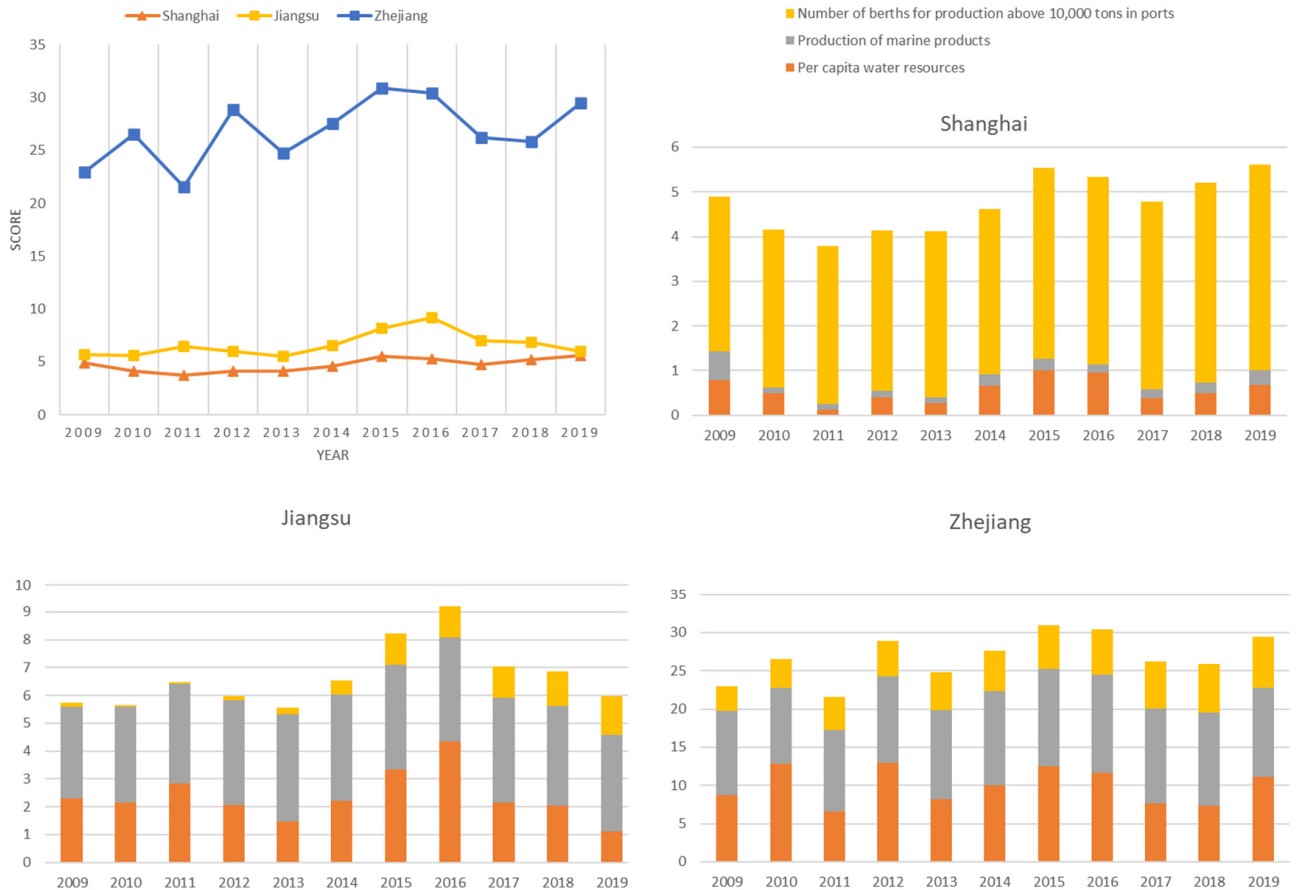

**Figure 2.** Yangtze River Delta marine resources indicator score from 2009 to 2019.

*4.5. Analysis of Technological Innovation*

Technological innovation is an important source of motivation for economic development [62]. Analyzed from the time evolution, Shanghai's score fluctuated significantly, while the other two provinces and cities fluctuated slightly upwards [63] (Figure 4). Shanghai's score increased rapidly to a very high value of 24.94 from 2009 to 2015 due to the increase in the output of marine research institutions, but the number of patents decreased significantly after 2016, causing the score to plummet to a very low value of 9.16 and rebound slowly thereafter. In Jiangsu Province, the number of marine researchers decreased significantly after 2015, and after 2016, with the increase in input and output, the score of Jiangsu Province rebounded and increased to a great value of 11.21 in 2018, and the score of Zhejiang Province decreased in 2016 due to the decrease in the number of marine researchers and scientific research institution papers and showed a rising trend thereafter. Analyzing regional differences, Shanghai relies on stronger financial advantages and superior geographical advantages to attract a large number of talents, which made Shanghai's score the greatest in the Yangtze River Delta region before 2015. However, in 2016, mainly

due to the decline in the number of marine researchers, thesis outputs, and the number of patent applications received, which led to a significant decrease in the overall scores of the three provinces, their gap also narrowed rapidly, and the level of science and technology innovation in the three regions maintained a basic parity level in 2018 and 2019. In 2017, Zhejiang Province overtook Jiangsu by a small margin due to the increase in its score in invention patents. Collectively, the provinces should continue to attract research talent continually and should place greater emphasis on research results, with a particular focus on increasing the number of patent applications.

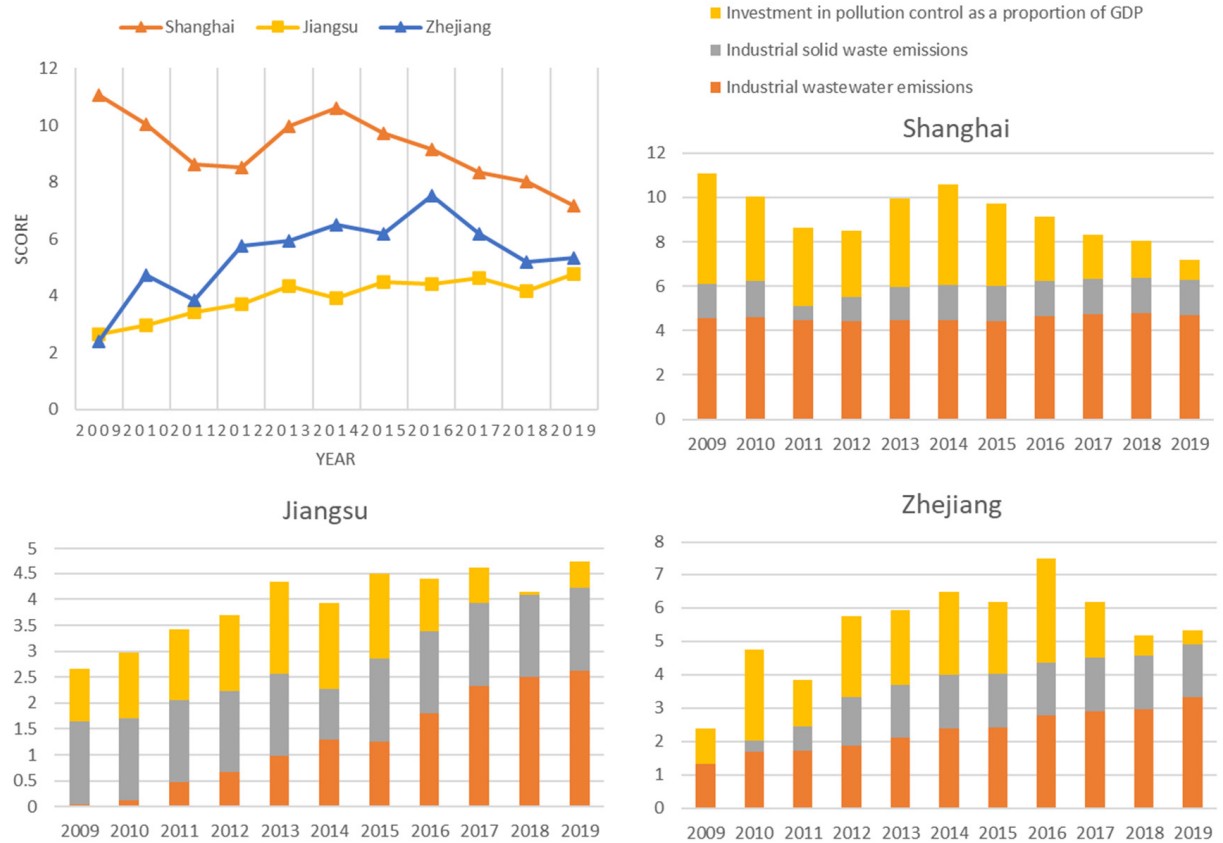

**Figure 3.** Yangtze River Delta ecological environment indicator score from 2009 to 2019.

### 4.6. Comprehensive Score Analysis

With the support of national development policies, the Yangtze River Delta region has been at the forefront of marine economic development in China [19]. In terms of time evolution, the level of marine sustainable development in the Yangtze River Delta provinces fluctuated and rose (Figure 5). Among them, Shanghai's score fluctuated significantly, while the other two provinces and cities listed a relatively stable trend.

Relying on its larger industrial scale and talent advantage, Shanghai vigorously developed the marine tertiary industry, breaking through the limitation of resources through the adjustment of industrial structure and transforming into an intensive and efficient marine economic development mode. The level of sustainable development of the marine economy rose steadily from 2009 to 2015, much higher than the regional average, and the score grew rapidly to a great value of 59.58 but plummeted to a very small value of 44.07 in 2017 by the weakness of scientific and technological innovation drive, and has been in a slow rebound since then. Zhejiang relied on rich water resources and marine biological resources to fully promote the growth of the marine economy. The overall level of sustainable development of the marine economy in Zhejiang Province showed a fluctuating upward trend and surpassed Shanghai in 2016, and has been in first place in the overall score for four years since then. Jiangsu Province took its rich marine resources and high marine science and

technology innovation capacity as a good foundation for its marine economy development. However, the scale of Jiangsu's marine economy compared to the region was small and the level of sustainable development of the marine economy was relatively backward, always below the regional average level and in the process of slow growth.

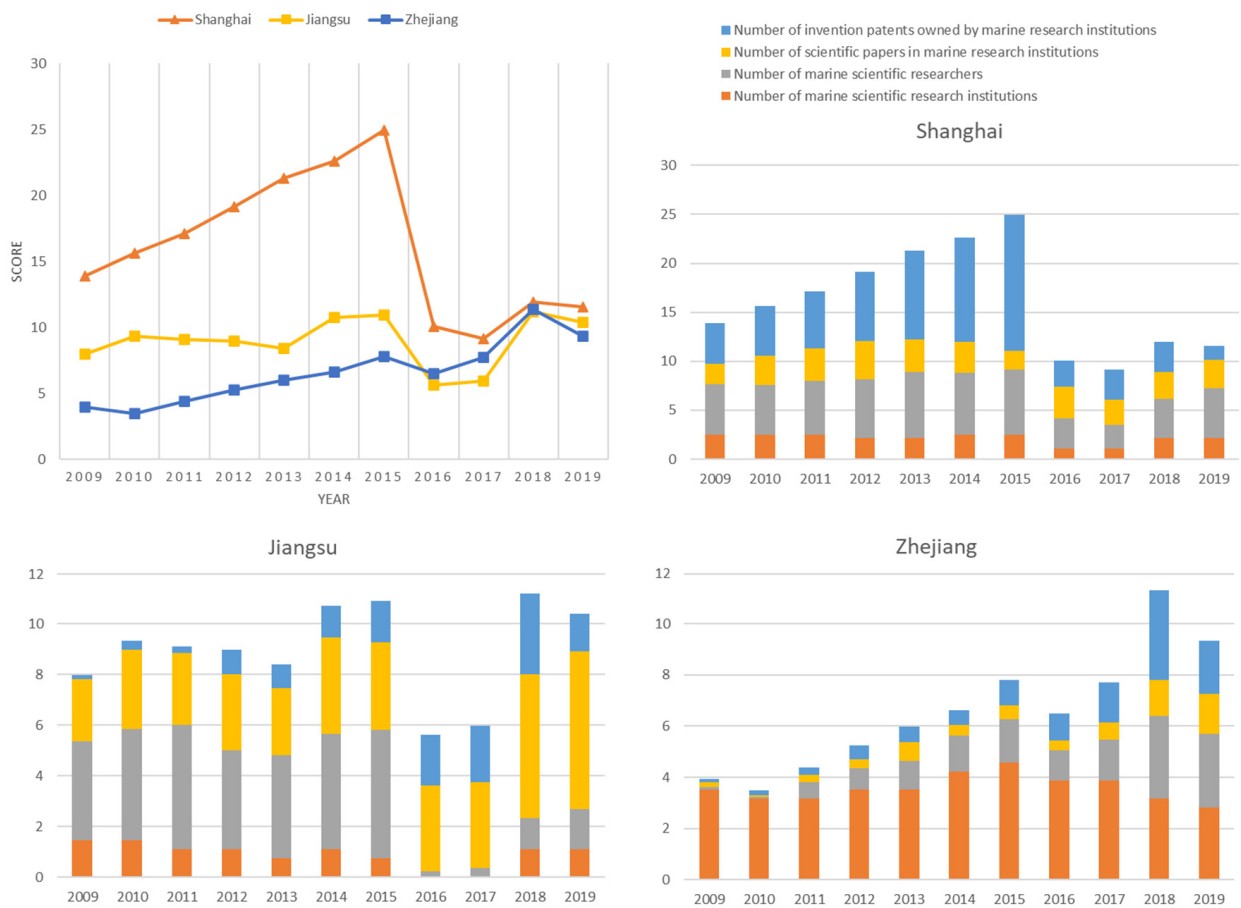

**Figure 4.** Yangtze River Delta science and technology innovation indicator score from 2009 to 2019.

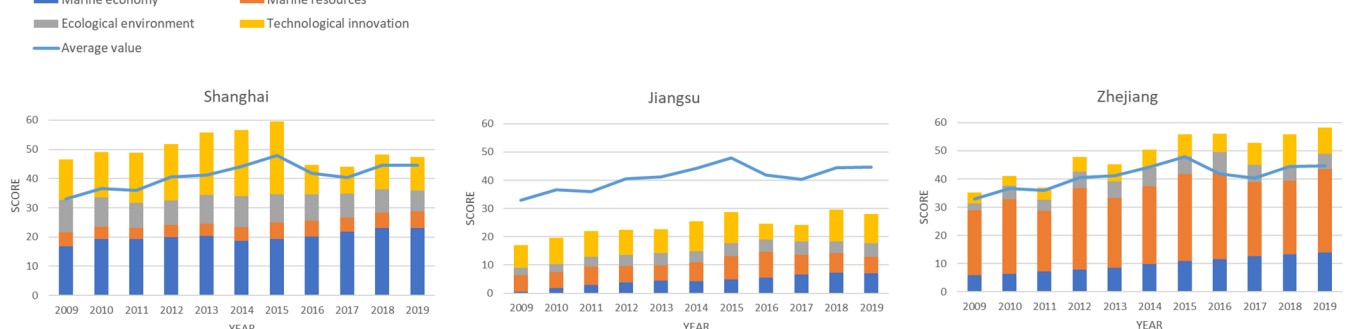

**Figure 5.** Yangtze River Delta comprehensive score from 2009 to 2019.

## 5. Discussion

Based on the theory of sustainable development, which is a fundamental strategy to guide social and economic transformation [64], this paper assessed the level of the sustainable development of the regional marine economy in four dimensions: the current situation of marine economy, environment, resources, and science and technology innovation, aiming to provide strategic suggestions to promote the sustainable development of regional marine economy and reduce regional differences. Sustainable development is a key task for coastal countries to gain a competitive advantage in the marine economy in the long

term, and existing studies have revealed the importance of sustainable development of the marine economy [65] and the driving role of various factors on sustainable development.

It has been shown that over-dependence on marine resources is not conducive to the sustainable development of the marine economy [66]. This study argues that the decisive role of marine resources in the development of the marine economy at this stage is related to the stage of the sustainable development of China's coastal areas [67]. This suggests that the establishment of marine economic development policies should take into account the current situation of each region's development and not directly borrow from the experience of other countries.

In this study, we believe that the current situation of the marine economy becomes an important support force for the sustainable and stable growth of the marine economy [68], and the quality, scale, and industrial structure of the marine economy will affect its sustainable level. In addition, the level of marine ecological environment also forms a constraint to sustainable development [69], and the government's attention to environmental pollution is one of the important foundations of sustainable development [70].

Existing empirical studies show that science and technology innovation has a driving effect on sustainable development [71]. This paper applies the viewpoint to the field of marine economic development and analyzes the relationship between marine science and technology innovation and the sustainable development the of marine economy. The results show that marine science and technology innovation is the intrinsic driving force of the sustainable development of the marine economy [72], but the conversion rate of regional scientific research results is not high at this stage, and the contribution rate of science and technology innovation to the marine economy needs to be improved.

At the same time, there are some limitations in this study. (1) This study did not use the most recent data and failed to measure and study the recent situation of the marine economy. (2) Due to the wide scope of the marine economy, some factors that are useful for measuring the sustainability of the marine economy were ignored. Therefore, future studies may update the data and establish a more complete evaluation system; in addition, with the growing recognition of the importance and necessity of the synergistic development of the marine economy and the environment, sustainable solutions for the marine economy can be further developed.

## 6. Conclusions and Recommendations

### 6.1. Conclusions

Grounded on the understanding of marine economic sustainable development, we constructed an evaluation index system for marine economic sustainable development from four dimensions—marine economy, marine resources, ecological environment, and technological innovation—and measured the level of marine economic sustainable development in the three provinces of the Yangtze River Delta from 2009 to 2019. The main conclusions are as follows: (1) There were significant differences in the industrial structure of the marine economy among regions in the Yangtze River Delta. (2) There was a large gap in the allocation and utilization of marine resources among provinces and cities in the Yangtze River Delta. In terms of the development, utilization, protection, and comprehensive management of marine resources, there was a lack of coordination and control capacity among regions, and there is a communication mechanism deficiency in cross-regional maritime resource circulation and maritime project construction. (3) The discharge of wastewater and waste in the Yangtze River Delta region was significantly controlled, but the investment in marine environmental governance was insufficient, seriously affecting the sustainable development of the marine economy in the region. (4) The support capability of marine technological innovation was insufficient. In terms of investment in technological innovation, the needs of marine researchers need to be further increased. In terms of technological innovation output, the conversion rate of scientific and technological achievements was relatively low, and there was a lack of core competitiveness in the production of marine technology patents.

*6.2. Suggestions for Countermeasures*

6.2.1. Strengthen Cooperation and Promote Regional Collaborative Development

The Yangtze River Delta is the intersection of "One Belt, One Road" and the Yangtze River Economic Belt and plays a leading role in China's economic and social development. The Yangtze River Delta region has played a leading role in China's economic and social development. The integrated development of the Yangtze River Delta provides opportunities for regional marine industry and economic as well as marine development; that is, collaborative innovation, attracting more international and domestic industrial and innovation resources to the Yangtze River Delta region. Based on the interconnection of infrastructure in the Yangtze River Delta region, provinces and cities strengthen regional cooperation to accelerate the flow of resources and shrink regional differences in marine resources.

6.2.2. Improve the Innovation Power of Marine Science and Technology and Improve the Efficiency of Transforming Marine Science and Technology Achievements

The rapid economic development of the Yangtze River Delta region has, to a certain extent, given it the advantage of attracting scientific research talents and applying for major marine research projects, but the lack of scientific research innovation output capacity has weakened the driving force of marine science and technology for economic development. Therefore, to address the problem of R&D investment in marine science and technology, the government should pay attention to the R&D of marine science and technology, increase the investment in science and technology innovation, orientate the training of relevant talents, and continuously increase the number of marine research personnel and institutions. For the problem of R&D output, through capital investment, promote the development of the traditional marine industry to a high-end direction and promote the transformation of scientific research results into actual productivity. Build marine enterprise brands, enhance the international visibility of enterprises, attract foreign investment, and improve the efficiency of the transformation of marine science and technology achievements. Through marine science and technology innovation, solve the technical problems faced by emerging industries such as the marine biomedical industry and high-end engineering equipment manufacturing, accelerate the improvement of the industrial structure of the marine economy, and promote the marine economy to achieve more efficient development.

6.2.3. Improve the Relevant System of Marine Environmental Protection and Increase the Protection of the Ecological Environment

The destruction of the marine ecological environment seriously restricts the development of major marine economic industries such as marine fishery, marine tourism, and marine biomedicine and hinders the growth trend of the marine economy. Therefore, the region should continue to strengthen the comprehensive management of marine pollution, increase investment in marine environmental management, continue to strictly control the discharge of pollutants such as solid waste and wastewater into the sea, and build ecosystem restoration projects to ecologically restore the heavily polluted marine areas. Detailed ecological and environmental protection policies have been developed to promote marine economic development at a pace that takes into account the preservation of the ecological environment. In order to better cope with the environmental dilemma, the world should work together to coordinate the planning of marine ecological protection in order to promote the safer development of the marine economy.

**Author Contributions:** X.N. and Y.Q. designed the original ideas presented in this manuscript. X.N. collected the relevant data. X.N. worked on data analysis. Y.Q. and X.N. wrote the original manuscript draft. Y.Q. participated in manuscript preparation and improvement. All authors have read and agreed to the published version of the manuscript.

**Funding:** The work was funded by the National Social Science Foundation for "Building a Strong Marine Power in the New Era" (Grant No. 22VHQ010) and the Zhejiang Provincial Cultural Research Project: "Exploration and Practice of Marine Economic Development for Common Prosperity—Putuo Case in Zhoushan, Zhejiang" (Grant No. 21WH70098-20Z).

**Institutional Review Board Statement:** Not applicable.

**Informed Consent Statement:** Not applicable.

**Data Availability Statement:** The datasets analyzed for this study can be found in the China Marine Economic Statistics Yearbook, China Environmental Statistics Yearbook, and provincial local yearbooks.

**Conflicts of Interest:** The authors declare no conflict of interest.

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
