# Peer review of "Measuring the Sustainable Development of Marine Economy Based on the Entropy Value Method: A Case Study in the Yangtze River Delta, China"

_sustainability, doi:10.3390/su15086719_

Round 1
Reviewer 1 Report
Delighted to read your manuscript that focuses on the Sustainable Development of Marine Economy. Recent years have witnessed a phenomenon that the importance of Sustainable Development of Marine Economy enjoys a rapid growth in the Yangtze River Delta. Therefore, it is high time that we should take it into consideration. After reading your manuscript, I put forward some following issues.
1. The focus of this manuscript is not accurate enough. Focusing on analysing and solving the specific problems of the Marine Economy in the Yangtze River Delta rather than emphasizing on weighing the evaluation indicators will be better.
2. The innovation of this manuscript is not prominent enough compared to other articles in the same fields.
3. The established evaluation index system of Sustainable Development of Marine Economy is not comprehensive enough.
4. Without coordinating the relationship between the primary indicators, the evaluation index system is considered incomplete.
5. The unclear definition of the indicator range leads to a lack of the rationality and scientificity of the data.
6. The manuscript does not take the negative impact of the development of the tertiary industry on marine ecology into account.
7. The spelling of this manuscript requires proofreading and I recommend English language editing by a native speaker.
Reviewer 2 Report
Dear author(s)
In order to identify the factors necessary for the sustainability of the maritime economy, this study has developed a new statistical index, analysed it using statistical data and compared it in three regions of China. The debate on the blue economy has only just begun, and the identification of relevant actors and their interrelationships await further analysis. In this context, this study will provide useful insights. However, there are several points that need to be improved in order for this study to be published.
Major comments:
The first is to clarify the research question in the introduction. A number of previous studies are mentioned in this section. However, it does not discuss what the limitations of those previous studies are and how this study overcomes those limitations. Besides, it mentions previous studies of sustainability indicators, but does not point out what shortcomings those indicators suffer from. Therefore, it does not explain how the measurement by entropy values, which this study introduces, ameliorates these shortcomings. These two points make the significance of the present study with respect to previous studies unclear. This point should be made more explicit.
Second, the sources of the data should be explained more carefully. Not only should the data sources be listed, but it should also be stated which data were extracted from which data sources.
Thirdly, in Table 2, the indicators developed on the basis of the results of the panel data are listed. However, no mention is made of the significance levels corresponding to the results of the panel data. Were all results statistically significant? Alternatively, have these indicators been developed by utilising results that are not statistically significant? This is important for the reliability of the results and should be properly explained. In the first place, there is also no explanation of what regression equations were estimated in the panel data analysis. It is therefore difficult to judge or verify the validity of the results in Table 2.
Minor comments:
The "where" in lines 239 and 243 does not need to be indented. Correct other similar points.
Lines 264, 295, 335, etc. should be left blank to allow for spacing between figures and tables. The same applies in other places.
Author Response
Major comment 1: The first is to clarify the research question in the introduction. A number of previous studies are mentioned in this section. However, it does not discuss what the limitations of those previous studies are and how this study overcomes those limitations. Besides, it mentions previous studies of sustainability indicators, but does not point out what shortcomings those indicators suffer from. Therefore, it does not explain how the measurement by entropy values, which this study introduces, ameliorates these shortcomings. These two points make the significance of the present study with respect to previous studies unclear. This point should be made more explicit.
Response: Thank you for your suggestion. We have revised the introduction based on your comments. (Line146-151)
Major comment 2: Second, the sources of the data should be explained more carefully. Not only should the data sources be listed, but it should also be stated which data were extracted from which data sources.
Response: In the data sources section, we have added specific sources and links to the data (Line 259-266).
Major comment 3: Thirdly, in Table 2, the indicators developed on the basis of the results of the panel data are listed. However, no mention is made of the significance levels corresponding to the results of the panel data. Were all results statistically significant? Alternatively, have these indicators been developed by utilising results that are not statistically significant? This is important for the reliability of the results and should be properly explained. In the first place, there is also no explanation of what regression equations were estimated in the panel data analysis. It is therefore difficult to judge or verify the validity of the results in Table 2.
Response: Thank you very much for your suggestion. There is however no mention of significance testing of panel data weight indices in the existing papers that use the entropy method as a research method for panel data indicator systems. (Ref: Wang Q, Yuan X, Zhang J, et al. Assessment of the sustainable development capacity with the entropy weight coefficient method[J]. Sustainability, 2015, 7(10): 13542-13563.; He, Y, X, et al. Comprehensive evaluation of global clean energy development index based on the improved entropy method[J]. Ecological Indicators Integrating Monitoring Assessment & Management, 2018.) The entropy method is a method to determine the weights of indicators based on the magnitude of the information transmitted to the decision maker by each indicator. The greater the difference of a certain evaluation index, the smaller the entropy value, the more information the index carries, and the greater the corresponding weight. Entropy method as an objective assignment method can effectively avoid the influence of subjective factors and has a high reference value. Moreover, all statistical indicators are selected with reference to the existing policy background and existing research to ensure the scientific accuracy and reliability of the indicators. (Ref: Sun J, Miao J, Mu H, et al. Sustainable development in marine economy: Assessing carrying capacity of Shandong province in China[J]. Ocean & Coastal Management, 2022, 216: 105981.ï¼›Yan X, Shi X, Fang X. The internal dynamics and regional differences of China's marine economic evolution based on comprehensive evaluation[J]. Alexandria Engineering Journal, 2022, 61(10): 7571-7583.) Lastly, it is regrettable that our paper does not involve regression analysis of panel data, which we will explore in our future research.
Minor comments: The "where" in lines 239 and 243 does not need to be indented. Correct other similar points. Lines 264, 295, 335, etc. should be left blank to allow for spacing between figures and tables. The same applies in other places.
Response: Thanks for your careful checks. We are sorry for our carelessness. Based on your comments, we have made a correction to the format of the manuscript. We changed the indent (Line 289,293,295,298, 303) and set aside a gap between the figures and tables .

Reviewer 3 Report
The article is devoted to the study of Sustainable Development of Marine Economy. As an instrumental method, the authors use the Entropy Value Method.
After reading and reviewing this paper, I think it has potential for publication, but the authors should revise as comments below in order to improve the research soundness:
1. Table 1 shows indicators for assessing the sustainable development level, but the authors didn’t provide units of measurement. In addition, in our opinion, it is not worth adding to the Primary indicator "Marine economy" both "Gross marine economic product" and "Value added of major marine industries" at the same time, because Gross marine economic product includes Value added of major marine industries. This is one of the ways to calculate gross value added product.
2. In section “3.2 Data sources”, you need to add links to data sources
3. Formulas 1 and 2 contain ?_???, ?’_??? however, in lines 239 and 243 of the manuscript, the order of indexes i and j is confused: X_hji, Y_hji
4. The analysis in sections 4.2-4.6 should be expanded by comparing the obtained results with the results of other authors, as well as a more detailed explanation of the reasons for the decline/rise of the obtained indicators
5. In section “5.Discussion”, the authors indicate that “… this paper analyzes the relationship between the sustainable development of regional economies and the current economic situation, environment…”, but I didn’t see an analysis of the relationship in the article. The article only provides an analysis of the dynamics of various indicators.
Author Response
Comment 1: Table 1 shows indicators for assessing the sustainable development level, but the authors didn’t provide units of measurement. In addition, in our opinion, it is not worth adding to the Primary indicator "Marine economy" both "Gross marine economic product" and "Value added of major marine industries" at the same time, because Gross marine economic product includes Value added of major marine industries. This is one of the ways to calculate gross value added product.
Response: This a good suggestion and we have supplemented the measurement units for each indicator in Table 1. At the same time, due to the inclusion relationship between the "Gross marine economic product" and the " Value added of major marine industries " indicators, we have removed the " Gross marine economic product "from the "Marine Economic " indicators and recalculated the weighting score.
Comment 2: In section “3.2 Data sources”, you need to add links to data sources.
Response: Thank you for your suggestion, we have added links to the data sources section.
Comment 3: Formulas 1 and 2 contain ?_???, ?’_??? however, in lines 239 and 243 of the manuscript, the order of indexes i and j is confused: X_hji, Y_hji.
Response: We have carefully checked the paper and have modified the indicators involved in the formula. We again thank the reviewer for the question.
Comment 4: The analysis in sections 4.2-4.6 should be expanded by comparing the obtained results with the results of other authors, as well as a more detailed explanation of the reasons for the decline/rise of the obtained indicators.
Response: Thank you for your suggestion. We have revised the analysis section and added citations to the paper as you have commented.(Line342-540)
Comment 5: In section “5.Discussion”, the authors indicate that “… this paper analyzes the relationship between the sustainable development of regional economies and the current economic situation, environment…”, but I didn’t see an analysis of the relationship in the article. The article only provides an analysis of the dynamics of various indicators.
Response: Our previous statements are indeed misleading. We have made changes. (Line 560-563) Thank you again for your suggestion.

Round 2
Reviewer 1 Report
After the revision, the content of the literature review and discussion section of the article is more organized and the level of analysis is richer. However, the current version still has the following issues:
1.The model used in the article still lacks theoretical support.
2. Some content formats need to be rechecked, for example, the images in Line 411 are not labeled with their names.
Author Response
Response to Reviewer 1 Comments
Comment 1: The model used in the article still lacks theoretical support.
Response: Thank you for pointing out the issue. We have added relevant content on the theoretical basis of the method in section 3.3 Method. (Line276-288)
Comment 2: Some content formats need to be rechecked, for example, the images in Line 411 are not labeled with their names.
Response: Thanks for your careful checks. Based on your comments, we have made a correction to the format of the manuscript and adjusted the name of the images.
Reviewer 2 Report
Dear author.
This study has been appropriately revised in accordance with the peer-reviewed report. However, it is difficult to see the revisions in this coloured text. From next time onwards, coloured or bold text should be used to clearly identify the revised sections. I recommend that the Editor-in-Chief accepts this study for publication in the journal.
Sincerely.
Author Response
Comment: This study has been appropriately revised in accordance with the peer-reviewed report. However, it is difficult to see the revisions in this coloured text. From next time onwards, coloured or bold text should be used to clearly identify the revised sections. I recommend that the Editor-in-Chief accepts this study for publication in the journal.
Response: Thank you for your suggestion. Next time, we will use colored text to annotate the modified parts.
Reviewer 3 Report
All comments have been corrected
Author Response
Comment: All comments have been corrected
Response: Thank you for your comment.